# Effects of light spectra and [15]N pulses on growth, leaf morphology, physiology, and internal nitrogen cycling in *Quercus variabilis* Blume seedlings

**Jun Gao[1,2], Jinsong Zhang[1,2], Chunxia He[1,2] *, Qirui Wang[3]**

**1** Key Laboratory of Tree Breeding and Cultivation of National Forestry and Grassland Administration, Research Institute of Forestry, Chinese Academy of Forestry, Beijing, China, **2** Co-Innovation Center for Sustainable Forestry in Southern China, Nanjing Forestry University, Nanjing, China, **3** Henan Academy of Forestry, Zhengzhou, China

* hechunxia08@126.com

**Data Availability Statement:** All relevant data are within the paper and its Supporting information files.

## Abstract

Light spectra of sunlight transmittance can generate an interactive effect with deposited nitrogen (N) on regenerated plants across varied shading conditions. Total N content in understory plants can be accounted for by both exogeneous and endogenous sources of derived N, but knowledge about the response of inner N cycling to interactive light and N input effects is unclear. We conducted a bioassay on Chinese cork oak (*Quercus variabilis* Blume) seedlings subjected to five-month N pulsing with $^{15}NH_4Cl$ (10.39 atom %) at 120 mg $^{15}N$ plant$^{-1}$ under the blue (48.5% blue, 33.7% green, and 17.8% red), red (14.6% blue, 71.7% red, 13.7% green), and green (17.4% blue, 26.2% red, 56.4% green) lighting-spectra. Half of the seedlings were fed twice a week using a 250 ppm N solution with micro-nutrients, while the other half just received distilled water. Two factors showed no interaction and neither affected growth and morphology. Compared to the red-light spectrum, that in blue light increased chlorophyll and soluble protein contents and glutamine synthetase (GS) activity, root N concentration, and N derived from the pulses. The green-light spectrum induced more biomass allocation to roots and a higher percentage of N derived from internal reserves compared to the red-light spectrum. The $^{15}N$ pulses reduced the reliance on N remobilization from acorns but strengthened shoot biomass, chlorophyll content, GS activity, and N concentration. In conclusion, light spectrum imposed an independent force from external N pulse to modify the proportion of N derived from internal sources in total N content in juvenile *Q. variabilis*.

## Introduction

Anthropogenic activities have caused an increase in atmospheric nitrogen (N) deposition to forest ecosystems under climate change [1–3]. Intensive N input has altered the pattern of N cycling in tree populations [4], where sunlight spectra generated a combined influence with N

**Funding:** This research was funded by the Fundamental Research Funds for the Central Non-profit Research Institution of CAF (grant number CAFYBB2018ZB001).

**Competing interests:** The authors declare no conflict of interest.

deposition on understory regeneration. Empirical practices suggest a frequent photosynthetic photon flux density (PPFD) that was perceived by tree plants ranging within 60–80 µmol m$^{-2}$ s$^{-1}$ [5, 6]. This level of illumination was also found to fall in the range of understory sunlight transmittance [7]. Therefore, regenerated plants can come across the interactive influences of sunlight and N deposition at the understory layer. Along with the size of forest gaps, the interaction can vary by changing light-spectra and N pulsing levels. A knowledge gap still exists about combined light and N effects on understory forest regeneration.

Light is the most important environmental factor for plant growth and is a determinant of photosynthesis. The various responses of woody plants to different light spectra result from varied acclimations to lighting conditions of the understory layer [7]. Light from light-emitting diodes (LEDs) sources has been established as a solid and flexible tool for current tests on the response of juvenile trees to different spectra [5, 6, 8–15]. Responsive variables of tree seedlings are usually employed about growth and biomass parameters [5, 8, 14, 15], photosynthesis and stomatal conductance [5, 8, 15], and transplant performance [8, 14]. The change of lighting condition may result in dilution of nutrient concentration [8, 9, 12, 16]. This is caused by accelerated biomass accumulation in some given spectra while nutrients were not supplied to a required level [17]. Nutrient dilution greatly impairs seedling quality and negatively influences field performance [18, 19]. Therefore, the formation of nutrient dilution is one of the fatal factors that limit the success of forest regeneration.

A spectrum with high red light (600–700 nm) would induce a severer nutrition dilution symptom [6, 13]. The high blue-light LED spectrum (400–500 nm) could further accelerate biomass accumulation and stimulated the occurrence of nutrient dilution [14, 15]. The spectrum with high green light (500–600 nm) could not alleviate nutrient dilution for slowly growing species [11, 20]. Supplemental supply of exogeneous nutrient input can cope with nutrient dilution in juvenile trees subjected to some given lighting spectra [9, 12]. Detectable nutrient content, however, can also be derived from endogenous sources of nutrient reserves [21–24]. To our knowledge, current understanding about the light and N interaction on nutritional status of trees usually neglects inner source of derived N.

Oak spp. has a general episodic growth pattern with multiple flushes and new budbursts in a growing season. Their response to light spectra is full of uncertainties. For example, a spectrum enriched with red and far-red wavelengths promoted shoot growth in *Quercus ilex* in early growing stage, but the promotion shifted to be effective on roots in later stage [25]. Spectra that were high in green-light wavelength can benefit whole-plant quality of *Q. ithaburensis* var. *macrolepis* [8], but growth and biomass were unchanged in *Q. mongolica* Fisch. ex Ledeb no matter whether the green-light spectrum was involved [20]. Chinese cork oak (*Quercus variabilis* Blume) is a widely distributed deciduous broadleaf tree species in East Asia throughout temperate and subtropical regions (24˚ to 42˚ N and 96˚ to 140˚ E) [26, 27]. This oak is a valued hardwood species that can be used as a raw material for construction, furniture, and nortriterpenoid extraction [28]. Because of high N resorption but low growing speed [26], Chinese cork oak usually needs a long time as regenerated saplings in the understory layer. It has a high frequency to receive both transmitted sunlight and N deposition than other oak species [29]. Besides the external source of nitrogen supply, N demand by oak can also be met by remobilization from inner reserves and acorns, which can together decrease the reliance on exogeneous-source N to feed current growth. The natural trait of N remobilization also increases the complexity to distinguish external vs internal sources of derived N in the total content [30]. Uncertainties will continuously increase when exposed to the interaction of lighting spectra and exogeneous N pulse for Chinese cork oak.

In this study, Chinese cork oak seedlings was raised under controlled conditions, where different characteristics of light spectra were imposed with simulated N deposition. The

exogenous N input was labeled by [15]N-isotope to facilitate distinguishing inner or outside sources of derived N. The object was to quantify the amount and percent of N derived from the pulse (NDFP) vs N derived from reserves (NDFR) in response to combined N input and lighting spectra. We hypothesized that: (1) the blue-light spectrum would induce more N dilution than the red- and green-light spectra, and (2) N concentration would be diluted to decline unless meeting N pulse of deposition.

## Materials and methods

### Plant material and growing condition

Pre-germinated Chinese cork oak acorns were collected from a seed source in Southern Taihang Mountains (34˚58'-35˚4'N, 112˚24'-112˚32'E) in central China. Authors stated that the name of the authority who issued the permission for field acorn collection was Nanshan Forest Farm, Jiyuan City, Henan Province, China. Acorns were sterilized in K permanganate solution (0.5%, w/w) for 30 min and sown into 212 cm$^3$ (7 × 4 × 13 cm, top diameter × bottom diameter × height) volume trayed-cavities (4 × 8 individuals embedded in a tray) filled with N-poor substrates (Mashiro-Dust™, Zhiluntuowei A&F S&T Inc., Changchun, China). This growing medium was comprised of 70% peat, 10% spent mushroom residue (SMR), and 20% perlite. Chemical analysis revealed that this media contained 9.8 mg kg$^{-1}$ $NH_4$–N, 5.6 mg kg$^{-1}$ $NO_3$-N, 870.3 mg kg$^{-1}$ $PO_4$-P, pH of 5.3, with an electrical conductance (EC) of 0.86 dS m$^{-1}$. In mid-March, trayed acorns were incubated in the Laboratory of Combined Manipulations of Illumination and Fertility on Plant Growth (Zhilunpudao Agric. S&T Inc., Changchun, China) for 51 days until 80% germination. During germination, the indoor environment was controlled to 29/21˚C (day/night) and relative humidity to 58/61% (max/min). In early May, germinated seedlings with fine roots attached to the acorn were transplanted to pots (11.5 × 7.5 × 9.5 cm, top diameter × bottom diameter × height) with one individual in each pot. All seedlings for transplant were screened to a similar size to eliminate possible impact of initial variation. Pots were filled with the same growing medium as described above. Four pots were placed in one tank (40 × 60 cm, width × length). A total of 216 potted seedlings were placed in 54 tanks.

### LED lighting treatment

The tanked pots of seedlings were placed on iron shelves (2.0 × 0.5 ×1.5 m, height × width × length). Each shelf had three floors functioning as growing chambers (0.5 × 0.5 × 1.5 m, height × width × length). Two tanks were placed in each chamber so that each shelf contained six tanks with 24 seedlings. An LED panel (0.5 × 1.2 m, width × length; Pudao Photoelectricity, Zhiluntuowei A&F S&T., Inc., Changchun, China) was attached to the ceiling of each chamber. The LED panel provided lighting in a n18 h photoperiod, which has been proven to benefit the growth of slowly growing species [8, 31]. One hundred diodes were embedded in the downward surface of each panel at a spacing of 2 × 2 cm. Every diode was designed to emit blue, red, or green-light. The electric flow for diodes emitting the same type of light was controlled by one electrical administration transformer. Electric power for the array of red-light diodes was administrated by a 200 W transformer, while power for arrays of either green- or blue-lights diodes was administrated by a 135 W transformer. Changing the electric flow for one array of diodes could modify the photosynthetic photon flux rate (PPFD) intensity of the lights. Therefore, the spectrum of light that was emitted by one LED-panel was obtained by adjusting the electric flow for the other three diodes. The visual color of light from an LED panel is a mixture of wavelengths for blue, red, and green light at various PPFD intensities controlled by the electric flow.

Visually blue, red, and green lights were employed as mixed-wavelengths as visible lights from 400 to 700 nm (EVERFINE PLA-20, Yuanfang Elect. S&T Inc., Hangzhou, China) [32]. As our aim was to test the effect of different spectra, the intensity of PPFD was controlled to a similar level that produces ordinary growth [5, 14, 15] to eliminate unexpected error from the interaction between changes in lighting spectra and intensity [6, 9, 11–13]. The spectra for three visual colors of lights fulfilling the requirements above were generated as follows:

1. Blue-light spectrum: Electric flow was adjusted to 70%, 10%, and 10% of full power for arrayed diodes emitting blue, red, and green lights, respectively.

2. Red-light spectrum: Electric flow was adjusted to 10%, 30%, and 20% of full power for arrayed diodes emitting blue, red, and green-light, respectively.

3. Green-light spectrum: Electric flow was adjusted to 30%, 20%, and 100% of full power for arrayed diodes emitting blue, red, and green-light, respectively.

The outcome of the three spectra is shown in Fig 1 and the specific optical characteristics for the three spectra are provided in Table 1.

## $^{15}$N pulsing treatment

In this study, N was delivered to Chinese cork oak seedlings through pulsing at a rate of 19.6 kg N ha$^{-1}$ (~120 mg N plant$^{-1}$), which is the annual amount of wet N deposition for the natural population in the region of warm temperate forests [33]. This amount is equal to 117 mg N plant$^{-1}$. Four individuals were arranged in a 0.24 m$^2$ area using the N consumption of mature trees to pulse our seedlings. It was reported that 125 mg N plant$^{-1}$ can maximize dry mass production at a sufficient level of N input for oak (*Quercus ilex*) seedlings [34]. N was pulsed according to methods in previous studies [35, 36]. Seedlings were fed twice a week using a balanced nutrient solution [22] with 250 ppm N delivered through ammonium chloride ($NH_4Cl$). In short, the solution contained 60 ppm P and 100 ppm K with micro-nutrients added [22]. Exogenous N was labeled $^{15}$N as 10.39 atom % $^{15}NH_4Cl$ (Shanghai Research Institute of Chemical Industry, Yunling Road, Shanghai, China). A 10 mL nutrient solution was applied to the soil surface of each pot using a 5 mL pipettor and only soils were pulsed to avoid $^{15}$N contamination of shoots [35, 36]. N was pulsed over 40 applications in five months. Out of the random pair of tanked seedlings, only one tank of four potted seedlings received $^{15}$N pulsing, while the other tank of four seedlings received distilled water at the same volume as the control.

## Experiment design

This study was conducted as a split-block design with the main block being three LED spectra, each of which harbored two $^{15}$N pulse treatments. Three iron shelves with LED panels emitting blue, red, and green lights were assigned to one block, three randomly placed blocks of shelves were assigned as three replicates. Four seedlings in one tank (either with or without $^{15}$N pulse) per shelf floor were assigned as the basic unit of sampling and measurement. Three floors of tanked pots of seedlings were grouped to average the observations for combined spectrum and N treatment.

## Sampling and measurement

All four seedlings per tank were sampled and measured for height and root collar diameter (RCD) in mid-October. Four seedlings were assigned to two groups, with two randomly chosen seedlings per group. One group of seedlings were separated into shoots (leaves and woody

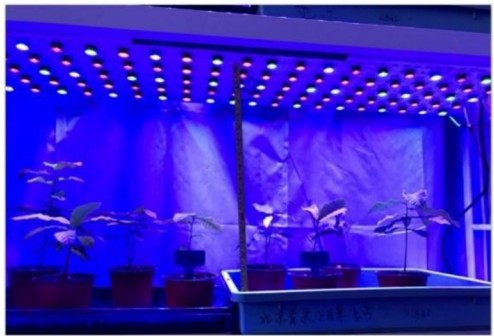

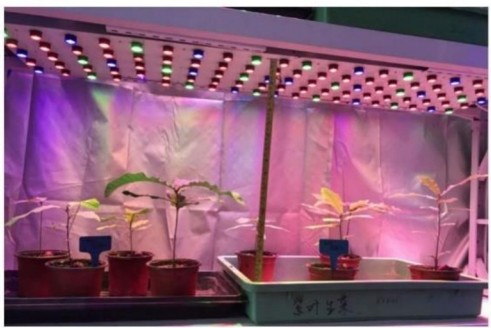

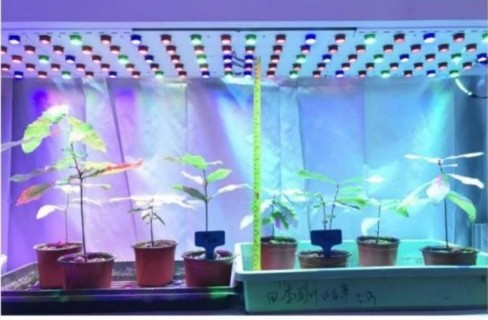

**Fig 1. Typical performance of Chinese cork oak (*Quercus variabilis* Blume) seedlings exposed to blue (48.5% blue, 33.7% green, and 17.8% red), red (14.6% blue, 71.7% red, and 13.7% green), and green (17.4% blue, 26.2% red, 56.4% green) colors of light-emitting diode (LED) spectra.** Tanks for potted seedlings were 40 × 60 cm. Green tanks contained seedlings subjected to [15]N pulses and black tanks contained seedlings subjected to water addition of the same volume.

**Table 1. Optical characteristics (40cm beneath the lighting source) of light-emitting diode (LED) lighting in blue, green, and red visible light colors of combined wavelengths in a wide bandwidth from 400 nm to 700 nm.**

| Optical characteristics | | Visible light color | | |
|---|---|---|---|---|
| | | Blue | Red | Green |
| PPFD [1] ($\mu$mol m$^{-2}$ s$^{-1}$) | | 95.18 | 97.86 | 96.46 |
| Proportional percent | Blue | 17.8% | 14.6% | 17.4% |
| | Red | 33.7% | 71.7% | 26.2% |
| | Green | 48.5% | 13.7% | 56.4% |

Note:

[1] PPFD, photosynthetic photon flux density.

stems), roots, and acorn. Roots were washed three times, by tap water once and distilled water twice, to carefully remove substrates without causing damage to fine roots. All three parts of the seedlings for each group were dried in oven at 65˚C to constant mass then weighed, ground, and measured for total N concentration and $^{15}$N enrichment using a stable isotope ratio mass spectrometer (Finnigan DELTA$^{plus}$ XP, Thermo Fisher Scientific, Grand Island, NY, USA) [37]. N content was calculated by the product of N concentration and biomass. Leaves of the other group of seedlings were directly used for determination or excised and stored in tinfoil-folded liquid N until measured for physiological parameters.

Four leaves were randomly chosen from two seedlings and scanned to obtain a 300 dots per inch (dpi) image (HP Deskjet 1510 scanner, HP Inc., Palo Alto, CA, USA) whose background was stratified and removed in Photoshop CS V. 8.0 (Adobe, San Jose, CA, USA). The front layer of leaves was opened as a histogram. The leaf green index (GI) can be directly read from the background data of the histogram [10, 38, 39]. Leaf area can be calculated as the total pixels of the histogram divided by the square of the dpi [10, 39, 40]. Scanned leaves were oven-dried at 65˚C to constant weight and measured for the biomass of a single leaf. Thereafter, the specific leaf area (SLA) was calculated as the product of leaf area and single leaf biomass.

Chlorophyll and protein concentrations were analyzed from fresh leaves using the method adapted from Zhao et al. [9]. In short, chlorophyll was determined using a 0.05 g sample, which was placed in a hydraulic bath at 65˚C for 1 h and used to determine concentrations of chlorophyll-a, chlorophyll-b, and carotenoid at wavelengths of 663, 645, and 470 nm, respectively. Protein concentration was measured from a 0.1 g sample that was ground in 1 mL of phosphate buffer at7.5 pH, centrifuged at 3000 rpm for 10 min, treated with 0.1 mL of Folin's reagent, and determined at 650 nm.

Foliar glutamine synthetase (GS) activity was assessed using the method reported by Wei et al. [40]. A 0.5 g leaf sample was homogenized in 5 mL extraction buffer (3.059 g Tris, 0.249 g MgSO$_4$·7H$_2$O, 0.309 g dithiothreitol, and 68.5 g sucrose dissolved in 500 mL deionized water brought to 8.0 pH using 0.05 mM HCl) at 6000 rpm for 20 min. We added 0.7 mL supernatant to 6 mL reaction B (6.118 g Tris, 9.959 g MgSO$_4$·7H$_2$O, 1.726 g monosodium glutamate, 1.211 g cysteine, 0.192 g Triethylene glycol diamine tetraacetic acid (EGTA), pH 7.4, 500 mL) using the solution of 0.7 mL ATP (40 mM). The mixture was incubated at 37˚C for 30 min and the reaction was stopped by adding 1.0 mL of ferric chloride reagent (3.317 g trichloroacetic acid, 10.102 g FeCl$_3$·6H$_2$O, 5 mL sulfuric acid, 100 mL). The absorbance of the product, glutamyl-γ-hydroxamate, was measured as 540 nm. Protein content was determined using the Folin's method as described above.

## Calculations

We calculated the isotopic abundance for N in atom % ($A_N$%) as [36, 41]:

$$A_N\% = \frac{^{15}N}{^{14}N + ^{15}N} \times 100. \tag{1}$$

The NDFP percentage was calculated as the relative specific allocation of pulse N ($RSA_N$%) as [24, 36, 42]:

$$RSA_N\% = \frac{A_{Pulse} - A_{Control}}{A_{Label} - A_{Standard}} \times 100, \tag{2}$$

where $A_{Pulse}$ is the $^{15}$N abundance in pulsed seedling organs (shoots, roots, and acorns), $A_{Control}$ is the $^{15}$N abundance in organs of controlled seedlings without pulsing, $A_{Label}$ is the $^{15}$N abundance in ($^{15}$NH$_4$)$_2$Cl solution, and $A_{Standard}$ is the ambient standard of 0.366% [24,

42]. The NDFP amount can be calculated by:

$$NDFP = RSA_N\% \times DM \times N_{Mass},$$ (3)

where DM is the dry mass of the seedling organ and $N_{Mass}$ is the N concentration in this organ. Therefore, the NDFR percentage can be calculated as 100% minus $RSA_N\%$ [36]. The NDFR amount can be calculated as [36]:

$$NDFR = (1 - RSA_N\%) \times DM \times N_{Mass}.$$ (4)

## Statistical analysis

Data were analyzed using SAS (ver. 9.4 64-bit, SAS Institute Inc., Cary, NC, USA). A Shapiro–Wilk test was conducted on data using the univariate procedure, logarithm transformation plus 1.0 was used for data with abnormal distribution to meet the normality requirement [43]. The effects of LED spectra (blue, red, and green) and $^{15}$N pulse (pulse vs control) were tested using two-way analysis of variance (ANOVA) employing a split-block model with the placement of three replicated blocks ($n = 3$) as the random factor in the mixed procedure. When a significant effect was indicated to be over-95% probability with degrees of freedom for the model and error of 5 and 12, respectively, means were arranged and compared according to Tukey's test at the 0.05 level.

## Results

### Leaf morphology and physiological indexes

The two treatments had no effect on LA or GI (Table 2), which ranged from 83.36 ± 7.03 to 107.55 ± 11.20 cm$^2$ and 86.49 ± 1.25 to 100.29 ± 16.92, respectively. The LED spectra treatment showed a significant effect on SLA (Table 2), which was higher in the red-light spectrum (512.84 ± 38.27 cm$^2$ g$^{-1}$) than in the blue (440.29 ± 45.50 cm$^2$ g$^{-1}$) and green (395.40 ± 21.83 cm$^2$ g$^{-1}$) light spectra.

Either lighting spectra or $^{15}$N pulse had a significant effect on chlorophyll-a and chlorophyll-b contents (Table 2). Both chlorophyll-a and chlorophyll-b contents were higher in the blue-light spectrum than in the red-light spectrum (Fig 2A and 2B). The $^{15}$N pulse increased the contents of both chlorophyll-a and chlorophyll-b relative to the control (Fig 2D and 2E). No effect was found on carotenoid content, which ranged between 0.15 and 0.25 mg g$^{-1}$ (Fig 2C and 2F).

The LED spectra had a significant effect on leaf protein content (Table 2), which was higher in the blue-light spectrum than spectra from red and green-light (Fig 3A). The $^{15}$N pulse did not have any impact on leaf protein content (Fig 3C). Both LED spectra and $^{15}$N pulses had a significant effect on GS activity (Table 2). Again, the blue-light spectrum resulted in higher GS activity than red and green spectra (Fig 3B). The $^{15}$N pulse also produced a significant increase in GS activity compared to the control (Fig 3D).

### Plant growth and biomass distribution

Neither LED spectra nor $^{15}$N pulse produced a significant effect on height or RCD (Table 2). Seedling height ranged between 16.70 ± 2.93 and 19.13 ± 1.91 cm, and RCD ranged from 0.35 ± 0.02 to 0.42 ± 0.05 cm. Both LED spectra and $^{15}$N pulse had a significant effect on biomass in shoot and root parts (Table 2). The green-light spectrum led to lower shoot biomass than the blue and red spectra (Fig 4A). In contrast, root biomass in the green-light spectrum

**Table 2.** *P*-values from analysis of variance (ANOVA) of lighting spectrum (LS), $^{15}$N pulse (NP), and their interaction (LS × NP) on growth and foliar physiology in Chinese cork oak (*Quercus variabilis* Blume) seedlings.

| Seedling parameter | Source of variation | | |
|---|---|---|---|
| | LS | NP | LS × NP |
| | *df* = 2 | *df* = 1 | *df* = 2 |
| LA | 0.6174 | 0.5320 | 0.5283 |
| GI | 0.0938 | 0.6977 | 0.8854 |
| SLA | 0.0004** | 0.0730 | 0.2901 |
| Chlorophyll-a content | 0.0241* | 0.0101* | 0.4547 |
| Chlorophyll-b content | 0.0144* | 0.0270* | 0.2196 |
| Carotenoid content | 0.1575 | 0.9566 | 0.5071 |
| Protein content | 0.0035** | 0.1346 | 0.2783 |
| Glutamine synthetase | <0.0001*** | 0.0228 | 0.7518 |
| Seedling height | 0.6139 | 0.6165 | 0.3341 |
| RCD | 0.2535 | 0.0824 | 0.1827 |
| Shoot biomass | 0.0067** | 0.0013** | 0.8486 |
| Root biomass | 0.0085** | 0.0037** | 0.9492 |
| Acorn biomass | 0.6218 | 0.0225* | 0.2192 |
| Shoot N concentration | 0.0164* | 0.0023** | 0.6068 |
| Root N concentration | 0.0023** | 0.0312* | 0.2200 |
| Acorn N concentration | 0.2118 | 0.0046** | 0.5194 |
| Shoot N content | 0.4390 | 0.0004** | 0.8771 |
| Root N content | 0.0020** | 0.8409 | 0.4896 |
| Acorn N content | 0.0134* | 0.0247* | 0.1168 |

Note: df, degree of freedom; LA, leaf area; GI, green index; SLA, specific leaf area; RCD, root-collar diameter; N, nitrogen; asterisks indicate levels of significant effects:

*, *P*<0.05;

**, *P*<0.01;

***, *P*<0.001.

was higher than in the other two spectra (Fig 4B). Compared with the control, the $^{15}$N pulse treatment resulted in larger shoot biomass (Fig 4D), but root biomass was lower (Fig 4E). The $^{15}$N pulse also lowered acorn biomass relative to the control (Fig 4F).

Both LED spectra and $^{15}$N pulse treatments had a significant effect on root to shoot biomass ratio (R/S) ($F_{5,12}$ = 21.51; $P$ < 0.0001). R/S was higher in the green-light spectrum (5.62 ± 1.66) than in the blue (3.25 ± 0.73) and red (3.06 ± 1.07) spectra. The $^{15}$N pulse lowered R/S by 41% relative to the control (R/S values: 2.97 ± 0.89 and 4.99 ± 1.57, respectively).

## Plant nitrogen content and distribution

Factors of LED spectra and $^{15}$N pulse treatments had a significant effect on N concentration in shoot and root parts (Table 2). Shoot N concentration was higher in the green-light spectrum than in the red-light spectrum (Fig 5A). However, root N concentration was higher in the blue-light spectrum than in the red-light spectrum (Fig 5B). The $^{15}$N pulse resulted in higher N concentration in both shoots (Fig 5E) and roots (Fig 5F) compared to the control. In contrast, the $^{15}$N pulse lowered acorn N concentration relative to the control (Fig 5F).

Although the LED spectra had no effect on shoot N content, the $^{15}$N pulse treatment showed a significant effect (Table 2). Compared to the control, the $^{15}$N pulse increased shoot N content by 74% (Fig 6D). The effect of LED spectra on root N content was significant (Table 2). The red-light spectrum resulted in lower root N content than the blue and green

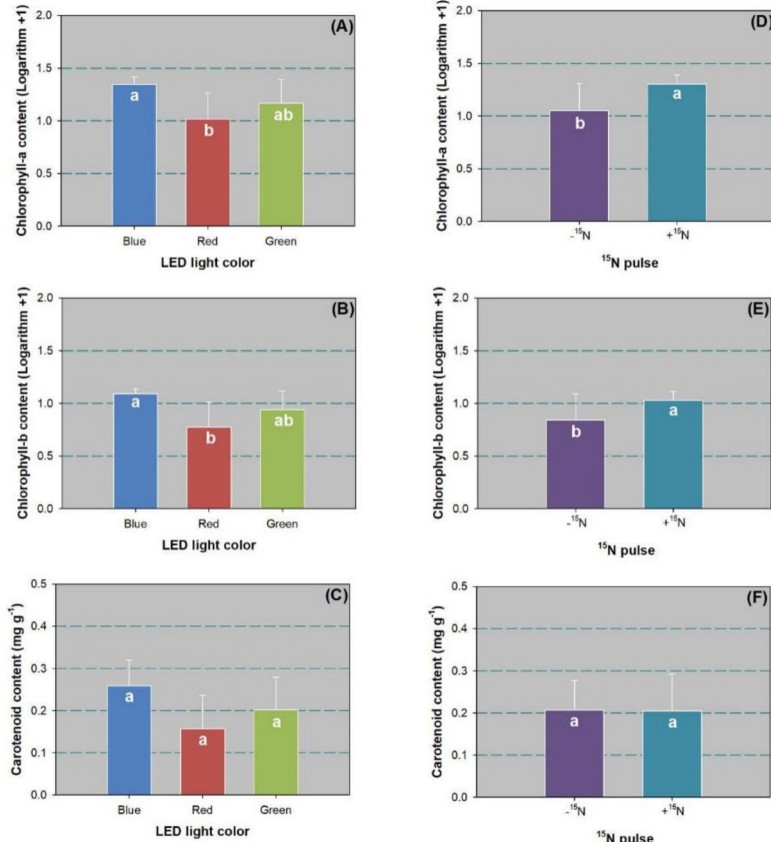

**Fig 2. Contents of chlorophyll-a (A,D), chlorophyll-b (B,E), and carotenoid (C,F) in leaves of Chinese cork oak (*Quercus variabilis* Blume) seedlings subjected to blue, red, and green LED spectra (*n* = 24 seedlings per replicate) or subjected to N pulse at 120 mg ¹⁵N plant⁻¹ (+¹⁵N) or zero (-¹⁵N) (*n* = 36).** Data that failed to follow a normal distribution were transformed by logarithm plus 1.0. Columns and bars represent means and standard errors, respectively. Different letters denote a significant difference according to Tukey's test at the 0.05 level.

spectra (Fig 6B). Acorn N content was higher in the green-light spectrum than in the blue-light spectrum by 34% (Fig 6C). The ¹⁵N pulse resulted in a decline in acorn N content by 17% (Fig 6F).

## Derived-nitrogen from internal and external sources

The LED spectra had no effect on the amounts of NDFP ($F_{2,6}$ = 1.60, $P$ = 0.2774) and NDFR ($F_{2,6}$ = 0.47, $P$ = 0.6482) in shoots (Fig 7A). The derived-N percentage in shoots was significant for both NDFP ($F_{2,6}$ = 7.75, $P$ = 0.0217) and NDFR ($F_{2,6}$ = 7.75, $P$ = 0.0217). Both NDFP and NDFR percentages in shoots were higher in the blue-light spectrum than in the red-light spectrum (Fig 7D).

The NDFP amount in roots was significantly affected by the LED spectra ($F_{2,6}$ = 49.41, $P$ = 0.0002). The red-light spectrum resulted in a lower NDFP amount than the other two spectra (Fig 7B). In contrast, the percent of NDFR in roots was highest in the red-light spectrum ($F_{2,6}$ = 10.93, $P$ = 0.0100), but the NDFP percentage was the lowest ($F_{2,6}$ = 10.93, $P$ = 0.0100; Fig 7E).

The NDFP amount in acorns was higher with the red-light spectrum treatment than under green-light ($F_{2,6}$ = 10.93, $P$ = 0.0100; Fig 7C). The percent of NDPF in acorns was also higher

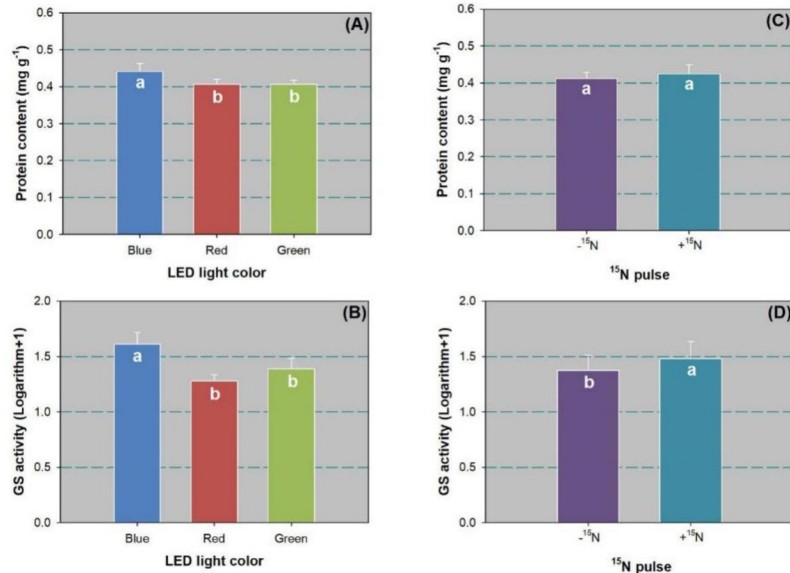

**Fig 3. Soluble protein content (A,C) and glutamine synthetase (GS) activity (B,D) in leaves of Chinese cork oak (*Quercus variabilis* Blume) seedlings subjected to blue, red, and green LED spectra lighting (*n* = 24) or subjected to N pulse at 120 mg $^{15}$N plant$^{-1}$ (+$^{15}$N) or zero (-$^{15}$N) (*n* = 36).** Data that failed to follow a normal distribution were transformed by logarithm plus 1.0. Columns and bars represent means and standard errors, respectively. Different letters mark significant difference according to Tukey's test at the 0.05 level.

in the red-light spectrum treatment ($F_{2,6}$ = 11.54, *P* = 0.0088; Fig 7F). In contrast, the acorn NDFR percentage was lower under the red-light spectrum than under green-light ($F_{2,6}$ = 11.54, *P* = 0.0088; Fig 7F).

## Discussion

The higher levels of chlorophyll and protein contents in leaves of Chinese cork oak seedlings in blue-light compared to the red-light treatment concur with findings in *Gerbera jamesonii* plantlets [44]. The synergetic increases of chlorophyll and protein appear to be a common response to up-regulating drivers that promote photosynthesis. For example, Li et al. [45] indicated that, in grape (*Vitis vinifera* L.), higher chlorophyll and protein contents resulted from the up-regulated genes in relation to microtubules, serine carboxypeptidase, and chlorophyll synthesis. These up-regulations also resulted in the down regulation of genes repressing protein. The blue-light spectrum was also found to up-regulated N assimilation and syntheses of photosynthetic pigments and ribulose-1,5-bisphosphate carboxylase oxygenase (Rubisco) protein in *Mesembryanthemum crystallinum* [46]. The GS activity was also enhanced, both in our study and in *P. koraiensis* seedlings [13]. Therefore, the blue light benefitted both syntheses of chlorophyll and protein by promoting N assimilation in Chinese cork oak. Above-mentioned results were accompanied by a high photosynthetic efficiency which were indicated by lower SLA in the blue light [47]. Overall, the blue- and red-lights resulted in contrasting responses of photosynthetic pigment and protein syntheses in Chinese cork oak seedlings. This can be explained by a summary that blue- and red-light generated a trade-off between the efficiencies of N-utilization and photosynthesis [48].

We failed to find any significant responses of seedling and leaf growth to either light spectra or $^{15}$N pulses for Chinese cork oak seedlings. LED spectra were also found to fail in imposing significant effect on LA in holm oak (*Q. ilex*) seedlings [25]. One study even reported a

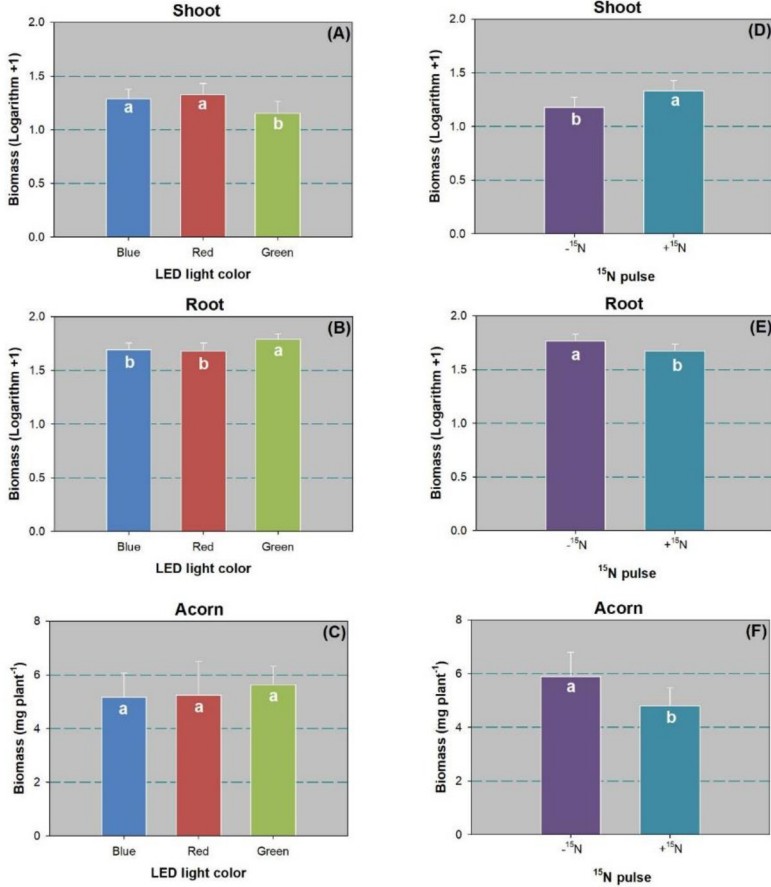

**Fig 4. Biomass in shoot (A,D), root (B,E), and acorn (C,F) of Chinese cork oak (*Quercus variabilis* Blume) seedlings subjected to blue, red, and green colors of LED spectra (*n* = 24) or subjected to N pulse at 120 mg [15]N plant[-1] (+[15]N) or zero (-[15]N) (*n* = 36).** Data that failed to follow a normal distribution were transformed by logarithm plus 1.0. Columns and bars represent means and standard errors, respectively. Different letters denote a significant difference according to Tukey's test at the 0.05 level.

negative effect of LED spectrum on holm oak seedlings [49]. In our study, Chinese cork oak seedlings may have suffered a significant difference of ontogeny among individuals that covered the effect from light or N pulse. The ontogeny may even generate a larger impact on seedling physiology than genotypic variation [50]. The episodic growth pattern of hardwood seedlings may be another factor that generates unexpected influence to hinder significant impacts [51]. Given that foliar physiology has been modified by different light spectra, it may need a longer term to reveal significant growth response.

Spectra from blue and red lights had a similar impact on biomass accumulation in all tissues. The lack of significant response of biomass production under different LED spectra in our study was also reported in former ones on *Q. ilex* [8, 52] and spruce [5, 15]. In contrast, the green-light spectrum induced greater biomass allocated to the root than the other two spectra. These results disagree with the common belief that green light may be not useful for plant growth because photons from green lights are reflected by green-color organs [53]. However, the green light has some physiological contributions to biomass production. A series of physiological activities are responsible for biomass response to the green-light spectrum, including but not limited to stomatal closure, plastid down-regulation, and restrictive P uptake

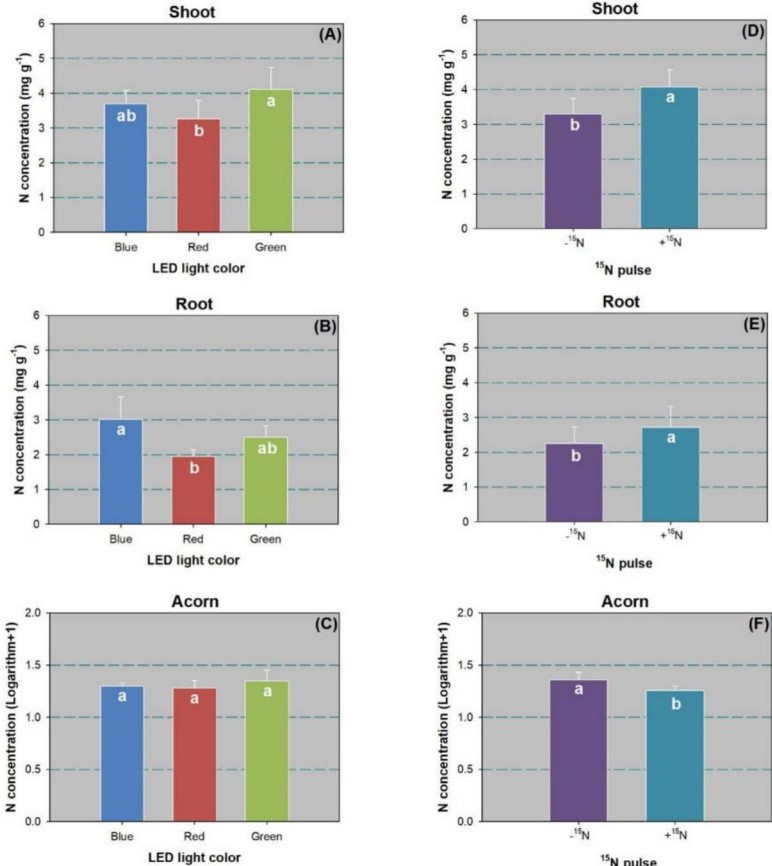

**Fig 5. Nitrogen (N) concentration in shoot (A,D), root (B,E), and acorn (C,F) of Chinese cork oak (*Quercus variabilis* Blume) seedlings subjected to blue, red, and green colors of LED spectra (*n* = 24) or subjected to N pulse at 120 mg $^{15}$N plant$^{-1}$ (+$^{15}$N) or zero (-$^{15}$N) (*n* = 36).** Columns and bars represent means and standard errors, respectively. Different letters denote a significant difference according to Tukey's test at the 0.05 level.

[10, 53]. The green light also brought promotion on biomass accumulation in *Chrysanthemum morifolium* plants [54] and *Plectranthus scutellarioides* cultivars [55].

Neither N content nor N concentration were not different in shoots subjected to blue vs red lights. However, red light induced high percent of derived N from internal source than blue light and inverse results for external N. These suggest that the red-light spectrum benefitted using internal N source while the blue-light spectrum benefitted the external N, although gross N allocation and accumulation were not differentiated. Interestingly, both N content and NDFP amount in roots were lower in the red-light spectrum than in the other two spectra but shoot N content was unchanged among the three spectra. As mentioned above, root biomass was not differentiated between the red- and blue-light treatments, but it was higher N concentration in roots in the blue light that resulted in another higher level of N content compared to the red-light treatment. These findings about root N content between blue- and red-lights concur with those on rice (*Oryza sativa* L.) seedlings [56], but do not agree with others on horticultural plants [10, 57]. Thus, low root N concentration in the red-light spectrum was also the result of restricted N uptake and assimilation [9, 11, 13]. Therefore, the red-light spectrum generated a restriction of external N uptake in roots without any driving impact on N allocation to aboveground organs. Higher levels of chlorophyl and protein contents and GS activities in leaves of seedlings subjected to the blue-light treatment was the result of promoting

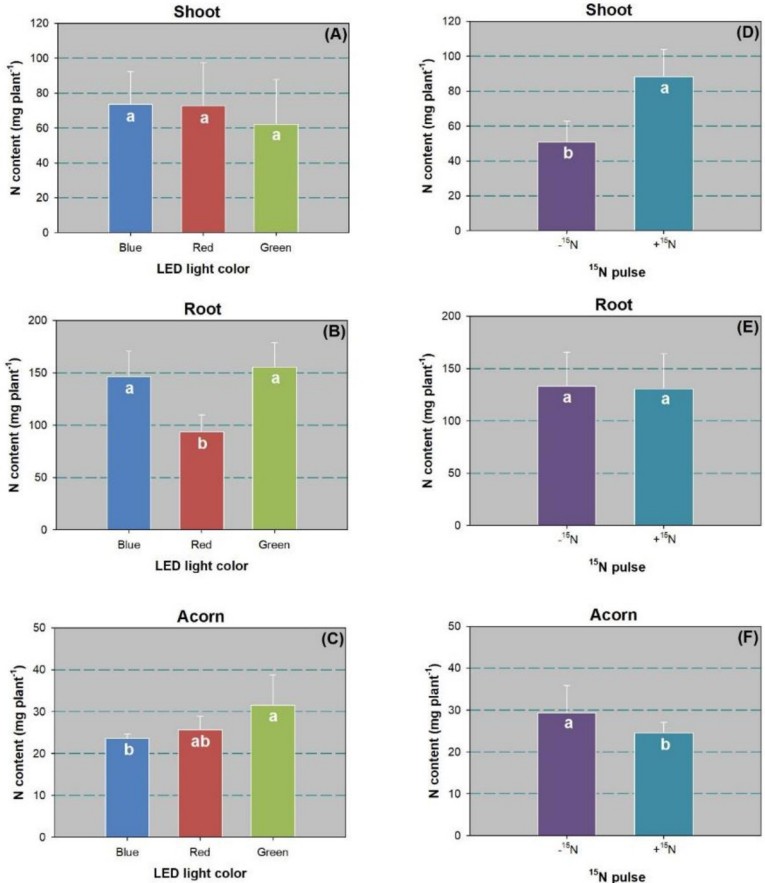

**Fig 6. Nitrogen (N) content in shoots (A,D), roots (B,E), and acorns (C,F) of Chinese cork oak (*Quercus variabilis* Blume) seedlings subjected to blue, red, and green LED spectra (*n* = 24) or subjected to N pulse at 120 mg [15]N plant[-1] (+[15]N) or zero (-[15]N) (*n* = 36).** Columns and bars represent means and standard errors, respectively. Different letters denote a significant difference according to Tukey's test at the 0.05 level.

synthesis by the blue light instead of N allocation. Overall, we cannot accept our first hypothesis because shoot N concentration was not lower in blue light than either red or green lights. The red light resulted in the most severe symptom of N dilution relative to green light due to lower N concentration with unchanged biomass in shoots.

Acorns did not show any responses in biomass or N concentration to LED spectra, indicating that lighting the above-ground organs has rare direct impact on belowground acorns. However, acorns subjected to green light presented higher NDFR percent but lower NDFP than those under red light. This is an acclimation to the green-light spectrum by acorns that derived more N from internal reserve when the external N adoption was limited compared to the red-light spectrum. We did not find similar reports that can be referred to for detecting the difference of acorns subjected to contrasting spectra. However, if green light can be taken as a kind of abiotic driver that can modify the source of derived N, our results can concur with those found in upland-forests where abiotic factors also imposed similar effect on acorns [58, 59]. The promotion to derive internal N by green light was accompanied by higher acorn N content in green light than in blue light. Neither biomass nor N concentration in acorns were different between the blue and green spectra, higher N content was the result of accumulative effect on errors in the ANOVA model of the product between biomass and concentration. As

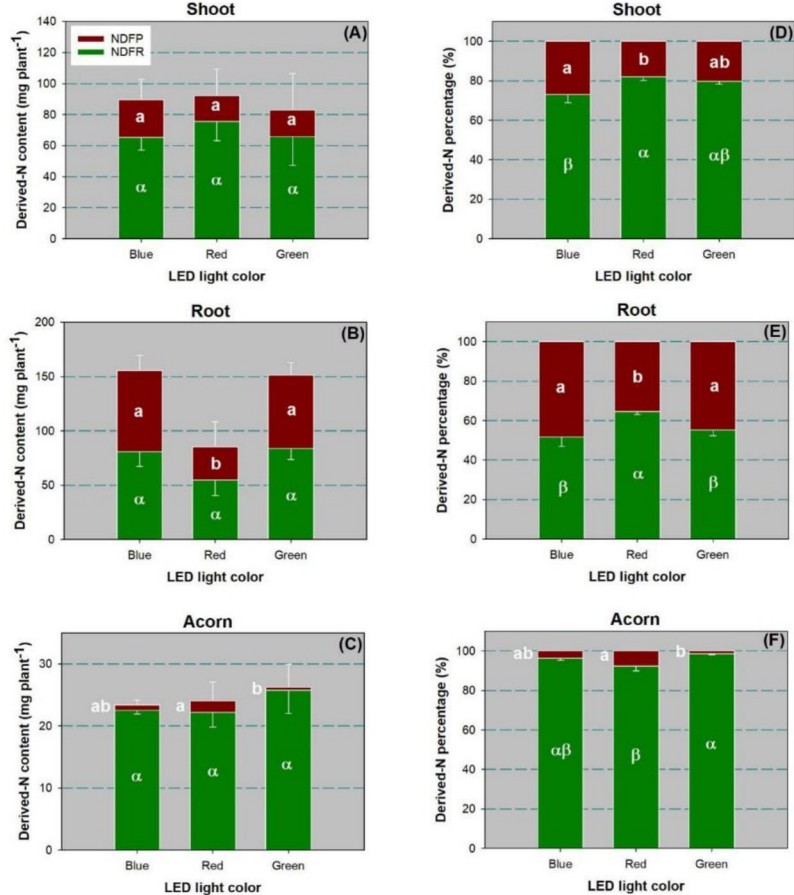

**Fig 7. Amount and percent of nitrogen (N) derived from pulses (NDFP) and plant reserves (NDFR) in shoots (A, D), roots (B,E), and acorns (C,F) of Chinese cork oak (*Quercus variabilis* Blume) seedlings subjected to blue, red, and green LED spectra (*n* = 24).** Columns and bars represent means and standard errors, respectively. Different letters denote a significant difference according to Tukey's test at the 0.05 level.

N content, both from internal and external sources, was all unchanged between the green- and blue-light spectra in shoot and root parts, lower N content in acorns in the blue light treatment cannot be explained by a stronger reliance on N remobilization from acorns.

We can accept our second hypothesis because N pulse did increase not only N concentration but also chlorophyll production, protein synthesis, and N assimilation. All of these are the result of accumulated N utilization under N deposition, which was also reported on red pine (*P. resinosa* Ait.) [60]. Declines in both N concentration and N content in acorns subjected to N pulse reflected reduced reliance on remobilization from acorns to feed other tissues when N availability is promoted [23]. This may also be related to the form of inorganic N input. We fed oak seedlings with $NH_4$-N as its assimilation with GS was sensitive to different spectra in *Fagus sylvatica* [49] and *P. koraiensis* seedlings [13]. Generally, oak prefers to uptake $NO_3$-N over $NH_4$-N from forest soils [61]. Our findings about the change in GS activities may also be the result of the associated response of nitrate reductase (NR) for $NO_3$-N assimilation since GS and NR showed a positive relationship with each other [13]. The plant component uses of SMR and peat may have involved some microbiomes that affected the $NH_4$-N and $NO_3$-N balance relative to the pure substrates in sands. However, N assimilation was affected by the N

amount trumping the mineral form, our results regarding N uptake and assimilation were not likely affected by these basic conditions.

## Conclusions

In this study, we tested internal N recycling in oak seedlings subjected to different lighting spectra. A bioassay simulating N deposition in *Quercus variabilis* Blume under blue, red, and green spectra and lights were provided by LED while seedlings were also subjected to $^{15}$N pulses. Different type of lighting spectra had varied functions on nutritional physiology of Chinese cork oak seedlings although no significant change can be seen by seedling morphology. The blue-light spectrum can benefit uptake and use of external N supply for photosynthesis and assimilation in leaves compared to the red-light spectrum. Also relative to the red-light spectrum, the green-light one promoted biomass allocation to belowground organs and benefitted the remobilization of internal N to the reserve. N deposition did not show any interaction with light spectra but benefitted N remobilization from acorns. Although we did not test far-red-light in our study, further work is suggested to detect the response of internal N recycling of oak or other tree seedlings exposed to the far-red spectrum. Future works are suggested to test the potential interaction between N deposition with lighting spectrum in a larger range of bandwidth.

## Supporting information

**S1 Raw data. Raw data have been supplied as supporting information file that can be disclosure from the link to this manuscript.**
(XLSX)

## Acknowledgments

Editors and reviewers who contribute to the improvement of this study.

## Author Contributions

**Conceptualization:** Jun Gao, Chunxia He.

**Data curation:** Jun Gao.

**Formal analysis:** Jun Gao, Chunxia He.

**Funding acquisition:** Jun Gao.

**Investigation:** Jun Gao.

**Methodology:** Jun Gao, Chunxia He.

**Project administration:** Jinsong Zhang.

**Resources:** Qirui Wang.

**Software:** Chunxia He.

**Supervision:** Jun Gao, Jinsong Zhang.

**Validation:** Jun Gao, Jinsong Zhang, Chunxia He, Qirui Wang.

**Visualization:** Jinsong Zhang, Chunxia He.

**Writing – original draft:** Jun Gao.

**Writing – review & editing:** Jun Gao, Chunxia He.

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
