## [Decision Letter · Decision Letter 0]

29 Jan 2021

PONE-D-20-37547

Light spectra modify nitrogen assimilation and nitrogen content in Quercus variabilis Blume seedling components: A bioassay with 15N pulses

PLOS ONE

Dear Dr. He,

Thank you for submitting your manuscript to PLOS ONE. After careful consideration, we feel that it has merit but does not fully meet PLOS ONE’s publication criteria as it currently stands. Therefore, we invite you to submit a revised version of the manuscript that addresses the points raised during the review process.

We look forward to receiving your revised manuscript.

Kind regards,

Xiao Guo, Ph.D.

Academic Editor

PLOS ONE

Additional Editor Comments:

As can be seen, both reviewers recommended major revisions. Therefore the manuscript cannot be accepted in its current condition and further revision is needed. The two reviewers gave excellent suggestions that will certainly improve your work (see below). Please response to each point of the comments one by one. Reviewer #2 mentioned that the language should be polished. I suggest the authors that the English language should be polished by a native English speaker or a professional company. A certificate of English editing by a professional company to guarantee the quality of the edit will be greatly appreciated.

Reviewers' comments:

Reviewer's Responses to Questions

**Comments to the Author**

1. Is the manuscript technically sound, and do the data support the conclusions?

Reviewer #1: Partly

Reviewer #2: Yes

2. Has the statistical analysis been performed appropriately and rigorously? 

Reviewer #1: Yes

Reviewer #2: Yes

3. Have the authors made all data underlying the findings in their manuscript fully available?

Reviewer #1: Yes

Reviewer #2: Yes

4. Is the manuscript presented in an intelligible fashion and written in standard English?

Reviewer #1: Yes

Reviewer #2: Yes

5. Review Comments to the Author

Reviewer #1: The manuscript has a novel title and rich indicators, which has a certain guiding significance for nitrogen efficient utilization of Quercus variables. However, there are some problems in writing：

75-77 The results of the above-mentioned studies demonstrate a knowledge gap that can be easily filled by new observations with studies using LED and N pulse to mimic natural conditions. What new observations?

86-90 What are the hypotheses based on before the experiment? It seems not mentioned in the introduction.

92 What is the test time (year, month)?

136 It can be seen from Figure 1 that the seedling growth (number of leaves, plant height) under the three visible light colors are not consistent, whether it will affect the test results? In the process of the experiment, should we choose the seedlings with the same growth status as far as possible?

218 There are many indicators in the result, but the writing is confusing. It is suggested to clarify the levels of the indexes from three aspects: 1. Leaf morphology and physiological indexes; 2. Plant growth and biomass distribution; 3. Plant nitrogen content and distribution. Meanwhile, it is suggested to use appropriate statistical methods to explain the relationship between indicators (such as correlation analysis).

138 What is the basis for such distribution from table 1？

267 What is the significance of analyzing N concentration and N content separately? If there is no specific significance, it is suggested to delete one section of the analysis.

250 It is suggested to combine table 2 and table 3 at the beginning of result.

290-316 Poor writing. Since there is no difference in the experiment, there is no need to spend a lot of time on it. The obvious experiment is meaningless. It is suggested that the discussion should be shortened. Moreover, the experimental data seems to be only one year, without repeated results for many years, which seems to be too hasty.

317-381 For the difference of seedling growth caused by different light color, the depth of discussion is not enough, and no more novel content is obtained. I think this is the place that needs to be improved. These discussions are not enough to explain the hypothetical problems. Previous studies should be cited from multiple perspectives (physiological, molecular) to explain the differences.

Reviewer #2: The authors studied the effects of light spectra (blue, red, and green spectra) and 15N pulses on Quercus variabilis Blume seeds. The idea is clear and the work is relative rich, but it is still far from publication and needs a major revision:

1. Please polish the language carefully.

2. The title of this manuscript may need to be changed, because you have also spent a lot of space describing the effects of light spectra and 15N pulses on other indicators of plants, such as Growth and Leaf Morphology, Leaf Physiology, Biomass Accumulation and Allocation.

3. The contents of Table 2 and Table 3 were basically expressed in the following figures. Do not express the same contents both in figures and tables, please delete or modify.

4. line 220-223: Here, these treatments have no significant effect on plant indexes, but the changes of indexes are mentioned later. Please rewrite to make it clear.

5. line 224: Maybe it should be "higher" instead of "lower"?

6. line 260: The "most" here is not rigorous, there are only two seedling tissue parts except for acorn. Please describe clearly.

7. line 276-279: This paragraph is obscure, please rewrite it.

8. line 282-284 and line 286-288: The sum of the percent of NDFP and the percent of NDFR is 100%, so if one is high, the other must be low. There is no need to repeat this known fact.

9. The Discussion section of this manuscript is very poorly written, please take the time to revise the discussion section carefully. The discussion section is used to discuss the important results of the study and their significance. I really don’t understand why the authors spent several paragraphs to write about the effects that they failed to find. You listed a bunch of insignificant examples (such as line 293-296) and made many unimportant literature comparisons, however, you did not discuss the important findings of this study in depth. This made the topic of this paper and the main scientific questions of the study were not clear.

10. line 323-325: please rewrite this sentense.

11. line 330: Should it be "not significantly" instead of "significantly"?

12. line 346-347: You said "another study" but didn't cite the literature.

13. line 356: Missing citation after "other studies".

6. PLOS authors have the option to publish the peer review history of their article (what does this mean?). If published, this will include your full peer review and any attached files.

Reviewer #1: No

Reviewer #2: No

---

## [Author Response · Author response to Decision Letter 0]

15 Mar 2021

5. Review Comments to the Author

Reviewer #1: The manuscript has a novel title and rich indicators, which has a certain guiding significance for nitrogen efficient utilization of Quercus variables. However, there are some problems in writing：

75-77 The results of the above-mentioned studies demonstrate a knowledge gap that can be easily filled by new observations with studies using LED and N pulse to mimic natural conditions. What new observations?

[Response] New observation refers to studies concerning the interaction between light spectra and N deposition in the understory condition. This term is not necessary and needs to be removed. 

86-90 What are the hypotheses based on before the experiment? It seems not mentioned in the introduction.

[Response] Generally the hypotheses should be put forward on the basis of current findings. It is naturally to expect to see the link between hypotheses to synthesized studies. However, the topic in this study that lighting spectra impact on inherent nutrient retranslocation has no basis that can be referred to. Please see the end of the paragraph that summarized the LED lighting on plant nutrition, it has been indicated that “… further explanation of the percent and amount of inner-derived nutrients by exposure to LED spectra is lacking” on Line 67 and 68. 

92 What is the test time (year, month)?

[Response] The study was commenced at early May and ended up in mid-October. All time courses have been added. 

136 It can be seen from Figure 1 that the seedling growth (number of leaves, plant height) under the three visible light colors are not consistent, whether it will affect the test results? In the process of the experiment, should we choose the seedlings with the same growth status as far as possible?

[Response] Yes, it is natural that hard wood seedlings have significant individual differences. This is due to the genetic difference from ontogeny for seedling variation. The episodic growth pattern will also contribute to the difference among individuals. This influence can be even more effective on physiology than genotype. All these natures have been well detected and established by the study group on hardwood regeneration in Purdue University. Please refer to one of their successional studies, Sloan and Jacobs (2016). The individual difference will generate some impact on results, not only in our study, but also for all studies if seedlings were taken as the materials. However, this type of impact belongs to the systematic error that can be avoided or reduced by statistics, replicates and means comparison. This is why some results fail to follow the expectation and show no difference. We cannot choose seedlings with uniform size or we will ruin the rule of statistics of random sampling. The screening for the even size has been done on the stage of transplant. Anyway, we have found some significant results that can support our hypotheses. This means that our statistics are fine to cope with systematic errors. However, we add something about this error to discussion. 

Reference:

Sloan, JL; Jacobs, DF. Ontogeny influences developmental physiology of post-transplant Quercus rubra seedlings more than genotype. Annals of Forest Science, 73(4): 987-993. 

218 There are many indicators in the result, but the writing is confusing. It is suggested to clarify the levels of the indexes from three aspects: 1. Leaf morphology and physiological indexes; 2. Plant growth and biomass distribution; 3. Plant nitrogen content and distribution. Meanwhile, it is suggested to use appropriate statistical methods to explain the relationship between indicators (such as correlation analysis).

[Response] Thanks the parameters have been re-classified again according to the suggested orders and sequences. We added a last section of ‘Derived-Nitrogen from Internal and External Sources’. We wished to use Pearson correlation to explain inter-relationship between variables at the initial stage of this article composition. However, we found we cannot achieve this analysis. This is because the core and most valuable meaning of this study was that we revealed internal and external N sources in oak seedlings subjected to different N and Light effects. Leaf, growth, biomass, and generally total N content variables were calculated for both controlled and N-deposited seedlings. However, derived N sources were calculated as the difference between deposition and control. This means that it is important to match N-source data to any other variables. 

138 What is the basis for such distribution from table 1？

[Response] As we explained in lines 123 to 126. These spectra were setting according to aims to enable ordinary growth for tree plants with uniform light photons, intensity, flux density, but just different spectra. We have listed eight citations that together support the basis of data in Table 1. 

267 What is the significance of analyzing N concentration and N content separately? If there is no specific significance, it is suggested to delete one section of the analysis.

[Response] N concentration, as it is shown in Figure 5, is the percentage of N amount on the unit weight of biomass. N content, shown in Figure 6, is the product of N concentration and biomass. Therefore, technically they are two different things and derived different sources of variables due to the involvement of biomass product. We cannot remove either of them because N concentration is the precondition to read results about N-derived percentage and N content about N-derived amount. 

250 It is suggested to combine table 2 and table 3 at the beginning of result.

[Response] These two tables have been combined together. 

290-316 Poor writing. Since there is no difference in the experiment, there is no need to spend a lot of time on it. The obvious experiment is meaningless. It is suggested that the discussion should be shortened. Moreover, the experimental data seems to be only one year, without repeated results for many years, which seems to be too hasty.

[Response] The first two paragraphs have been largely shortened. We did conduct a specific experiment with all environmental conditions controlled in the laboratory. We did not have to repeat our experiment in another as it is the necessary task for field studies. The necessity of cross-year experiment is needed for field investigation that would be largely impacted by unexpected episodes, such as extreme weather, unforeseen pest or bacterial infection, anthropogenic damage, and natural fires. Our study employed the classical methodology for a fully controlled experiment to detect the specific response of internal nitrogen cycling although sometimes nothing has been changed from outside. At least 20 studies have been published and obtained excellent citation records including those we refer to, such as Warren et al. [1], Salifu et al. [2], Wei et al. [3], Li et al. [4], He et al. [5]. 

References:

[1] Warren CR, Livingston NJ, Turpin DH. Response of Douglas-fir seedlings to a brief pulse of 15N-labeled nutrients. Tree Physiol. 2003; 23: 1193-1200.

[2] Salifu KF, Islam MA, Jacobs DF. Retranslocation, plant, and soil recovery of nitrogen-15 applied to bareroot black walnut seedlings. Commun Soil Sci Plant Anal. 2009; 40: 1408-1417.

[3] Wei HX, Xu CY, Ma LY, Wang WJ, Duan J, Jiang LN. Short-term nitrogen (N)-retranslocation within Larix olgensis seedlings is driven to increase by N-deposition: Evidence from a simulated N-15 experiment in Northeast China. Int J Agric Biol. 2014; 16: 1031-1040.

[4] Li XW, Chen QX, Lei HQ, Wang JW, Yang S, Wei HX. Nutrient uptake and utilization by fragrant rosewood (Dalbergia odorifera) seedlings cultured with oligosaccharide addition under different lighting spectra. Forests. 2018; 9: 15.

[5] He CX, Zhao Y, Zhang JS, Gao J. Chitosan oligosaccharide addition to Buddhist Pine (Podocarpus macrophyllus (Thunb) Sweet) under drought: Reponses in ecophysiology and delta C-13 abundance. Forests. 2020; 11: 13.

317-381 For the difference of seedling growth caused by different light color, the depth of discussion is not enough, and no more novel content is obtained. I think this is the place that needs to be improved. These discussions are not enough to explain the hypothetical problems. Previous studies should be cited from multiple perspectives (physiological, molecular) to explain the differences.

[Response] We have revised the discussion part. The novelty of our study is that we are the first to reveal derived-N sources in response to the combination of N deposition and light spectra. 

Reviewer #2: The authors studied the effects of light spectra (blue, red, and green spectra) and 15N pulses on Quercus variabilis Blume seeds. The idea is clear and the work is relative rich, but it is still far from publication and needs a major revision:

1. Please polish the language carefully.

[Response] After the revision of this round we will sent our manuscript to some professional agency to modify the language. 

2. The title of this manuscript may need to be changed, because you have also spent a lot of space describing the effects of light spectra and 15N pulses on other indicators of plants, such as Growth and Leaf Morphology, Leaf Physiology, Biomass Accumulation and Allocation.

[Response] We modified our title according to this suggestion. 

3. The contents of Table 2 and Table 3 were basically expressed in the following figures. Do not express the same contents both in figures and tables, please delete or modify.

[Response] I am sorry but data in Table 2 and 3 are P-values from ANOVA and according figures are results of difference which was preconditioned and ruled by Tables 2 and 3. However, we combined the two tables. 

4. line 220-223: Here, these treatments have no significant effect on plant indexes, but the changes of indexes are mentioned later. Please rewrite to make it clear.

[Response] This is an error. Amended. 

5. line 224: Maybe it should be "higher" instead of "lower"?

[Response] Yes it should be. Amended. Thanks.

6. line 260: The "most" here is not rigorous, there are only two seedling tissue parts except for acorn. Please describe clearly.

[Response] Thanks for correcting. Amended. 

7. line 276-279: This paragraph is obscure, please rewrite it.

[Response] It has been rewritten. 

8. line 282-284 and line 286-288: The sum of the percent of NDFP and the percent of NDFR is 100%, so if one is high, the other must be low. There is no need to repeat this known fact.

[Response] Thanks. The latter part has been removed. 

9. The Discussion section of this manuscript is very poorly written, please take the time to revise the discussion section carefully. The discussion section is used to discuss the important results of the study and their significance. I really don’t understand why the authors spent several paragraphs to write about the effects that they failed to find. You listed a bunch of insignificant examples (such as line 293-296) and made many unimportant literature comparisons, however, you did not discuss the important findings of this study in depth. This made the topic of this paper and the main scientific questions of the study were not clear.

[Response] The first two paragraphs have been largely shortened to avoid over too much discussion about unchanged results. We transferred the main attention in discussion to findings as the important part. 

10. line 323-325: please rewrite this sentense.

[Response] Rewritten. 

11. line 330: Should it be "not significantly" instead of "significantly"?

[Response] Thanks. This sentence has been revised. 

12. line 346-347: You said "another study" but didn't cite the literature.

[Response] Citation added. 

13. line 356: Missing citation after "other studies".

[Response] Other studies have been removed.

---

## [Decision Letter · Decision Letter 1]

5 May 2021

PONE-D-20-37547R1

Effects of light spectra and 15N pulses on growth, leaf morphology, physiology, and internal nitrogen cycling in Quercus variabilis Blume seedlings

PLOS ONE

Dear Dr. He,

Thank you for submitting your manuscript to PLOS ONE. After careful consideration, we feel that it has merit but does not fully meet PLOS ONE’s publication criteria as it currently stands. Therefore, we invite you to submit a revised version of the manuscript that addresses the points raised during the review process.

Dear authors, based on the revisions provided by three referees, I should inform that your manuscript cannot be accepted for publication in this current form and “major revisions” is needed.

Although reviewer 1 think that all comments have been fully addresses, reviewer 2 argued that the authors did not fully address his/her comments. As can be seen below, reviewer 2 pointed out the necessity to reorganize the introduction (to get the three hypotheses) and discussion. Reviewer 2 also complained about the language usage.

I kindly invite you to provide a revised version of your manuscript including the suggestions and comments of all reviewers, particularly the comments of reviewer 2. Together with your resubmission, please provide a point by point account of your revisions specifying how and where you addressed each suggestion. It is important not only to give answers in your response letter, but also to make the appropriate changes in the manuscript, wherever they are appropriate. Thus, my final decision will be taken after your revised manuscript has been reviewed by reviewers again.

In the previous response letter, the authors replied that “After the revision of this round we will sent our manuscript to some professional agency to modify the language”. I would strongly suggest that the authors polish the language before they submit the revised manuscript.

Best regards,

Xiao Guo

Academic Editor

We look forward to receiving your revised manuscript.

Kind regards,

Xiao Guo, Ph.D.

Academic Editor

PLOS ONE

Reviewers' comments:

Reviewer's Responses to Questions

**Comments to the Author**

1. If the authors have adequately addressed your comments raised in a previous round of review and you feel that this manuscript is now acceptable for publication, you may indicate that here to bypass the “Comments to the Author” section, enter your conflict of interest statement in the “Confidential to Editor” section, and submit your "Accept" recommendation.

Reviewer #1: (No Response)

Reviewer #2: (No Response)

Reviewer #3: All comments have been addressed

2. Is the manuscript technically sound, and do the data support the conclusions?

Reviewer #1: Yes

Reviewer #2: Yes

Reviewer #3: Yes

3. Has the statistical analysis been performed appropriately and rigorously? 

Reviewer #1: Yes

Reviewer #2: Yes

Reviewer #3: Yes

4. Have the authors made all data underlying the findings in their manuscript fully available?

Reviewer #1: Yes

Reviewer #2: No

Reviewer #3: Yes

5. Is the manuscript presented in an intelligible fashion and written in standard English?

Reviewer #1: Yes

Reviewer #2: No

Reviewer #3: Yes

6. Review Comments to the Author

Reviewer #1: (No Response)

Reviewer #2: The manuscript is still poorly written: the language is still irregular, and the statement is still unacademic.

The significance of this study needs to be summarized again. If the significance of this study only showed that red-light spectrum should be avoided for Q. variabilis regenerations, this study is too limited to be publish in this journal.

How did you get the hypotheses of this study? The three hypotheses were gotten without any reasoning.

The discussion section has not be improved. It is still not highlighting the significance of this study, no focus, no main line, logical confusion.

Reviewer #3: In this study, the authors examined the effects of different light spectra and 14 N pulses on morphology, physiology, and nitrogen (N) uptake performance of Quercus variabilis Blume seedlings. I think this manuscript was generally well designed and analyzed. But still there are some incorrect and awkward sentences that need to be double checked.

L54 Which classic model of the relationship?

L65 Is the effect of this mixed spectrum consistent with the single spectrum?

L71 Maybe the research background can be put in the first paragraph

L89, 90 What are the experimental hypotheses based on? It's not completely consistent with the results of others research (line 62-66). Maybe you can change it to questions.

L149 Only one tanked pot of seedlings received 15N pulsing? One in four? What is the number of total replications of seedlings received 15N pulsing?

L210-210 Paragraph spacing is different from before

L231 Please hang the first line

L270 Describe results that are significantly different firstly, rather than no significant

L310 But there was a significant influence of light on aboveground biomass in table 2

L 276-297: I suggest modifying these phrases. First describe what you found and then the possible explanation with citations, rather than a lot of citations. You can check that along all discussion.

Table 2 I suggest marking the P-values less than 0.05 with asterisks or letters

7. PLOS authors have the option to publish the peer review history of their article (what does this mean?). If published, this will include your full peer review and any attached files.

Reviewer #1: No

Reviewer #2: No

Reviewer #3: No

---

## [Author Response · Author response to Decision Letter 1]

24 May 2021

Responses to the reviewers: 

Thank you for submitting your manuscript to PLOS ONE. After careful consideration, we feel that it has merit but does not fully meet PLOS ONE’s publication criteria as it currently stands. Therefore, we invite you to submit a revised version of the manuscript that addresses the points raised during the review process.

[Response] Thanks for giving the revision decision. We will process of manuscript with suggested changes accordingly. 

Dear authors, based on the revisions provided by three referees, I should inform that your manuscript cannot be accepted for publication in this current form and “major revisions” is needed.

Although reviewer 1 think that all comments have been fully addresses, reviewer 2 argued that the authors did not fully address his/her comments. As can be seen below, reviewer 2 pointed out the necessity to reorganize the introduction (to get the three hypotheses) and discussion. Reviewer 2 also complained about the language usage.

[Response] We will specially pay attention to concerns and marks of reviewer 2 and revise the manuscript to the best extent to meet his/her requirements. 

I kindly invite you to provide a revised version of your manuscript including the suggestions and comments of all reviewers, particularly the comments of reviewer 2. Together with your resubmission, please provide a point by point account of your revisions specifying how and where you addressed each suggestion. It is important not only to give answers in your response letter, but also to make the appropriate changes in the manuscript, wherever they are appropriate. Thus, my final decision will be taken after your revised manuscript has been reviewed by reviewers again.

[Response] We will revise the manuscript according to comments and suggestions. 

In the previous response letter, the authors replied that “After the revision of this round we will sent our manuscript to some professional agency to modify the language”. I would strongly suggest that the authors polish the language before they submit the revised manuscript.

[Response] Thanks for remaindering. We will polish the language before revising the manuscript. 

Best regards,

Xiao Guo

Academic Editor

A rebuttal letter that responds to each point raised by the academic editor and reviewer(s). You should upload this letter as a separate file labeled 'Response to Reviewers'.

A marked-up copy of your manuscript that highlights changes made to the original version. You should upload this as a separate file labeled 'Revised Manuscript with Track Changes'.

An unmarked version of your revised paper without tracked changes. You should upload this as a separate file labeled 'Manuscript'.

We look forward to receiving your revised manuscript.

Kind regards,

Xiao Guo, Ph.D.

Academic Editor

PLOS ONE

Reviewers' comments:

Reviewer's Responses to Questions

Comments to the Author

Reviewer #1: (No Response)

Reviewer #2: The manuscript is still poorly written: the language is still irregular, and the statement is still unacademic.

[Response] We will polish our English language before revisions were made and promoted again after revision. 

The significance of this study needs to be summarized again. If the significance of this study only showed that red-light spectrum should be avoided for Q. variabilis regenerations, this study is too limited to be publish in this journal.

[Response] Significance has been reconsidered and revised with a new expression in Abstract and Introduction parts. Current significance in scientific contribution reads limited because it concerns the results on species in Abstract. Therefore, Abstract will be revised to reveal results about deep mechanism for scientific contributions. Introduction and Discussion will be revised accordingly. The scientific contribution of this study was to test the modification of internal vs outside sources of derived N that was conducted on Q. variabilis seedlings. 

How did you get the hypotheses of this study? The three hypotheses were gotten without any reasoning.

[Response] Hypotheses have all been replaced by new ones that are supported by cited studies. 

The discussion section has not been improved. It is still not highlighting the significance of this study, no focus, no main line, logical confusion.

[Response] The whole discussion part has been totally revised. 

Reviewer #3: In this study, the authors examined the effects of different light spectra and 14 N pulses on morphology, physiology, and nitrogen (N) uptake performance of Quercus variabilis Blume seedlings. I think this manuscript was generally well designed and analyzed. But still there are some incorrect and awkward sentences that need to be double checked.

[Response] Thanks for indicating flaws of our manuscript. We will revise the sentences and double checked them accordingly with the revision by marks of other reviewers and editors. 

L54 Which classic model of the relationship?

[Response] The classic model is the exponential fertilization model that was formally put forward by Timmer in 1997. Here one explanation was lacked to indicate that, in this model, dilution status of nutrients in seedlings was caused by accelerated biomass accumulation without fertilizer supply. The biomass accumulation can be promoted by a prolonged photoperiod, but fertilizer regime was not changed, which together resulted in the dilution. We revised this part. 

L65 Is the effect of this mixed spectrum consistent with the single spectrum?

[Response] In the range of cited studies, we can say yes. 

L71 Maybe the research background can be put in the first paragraph

[Response] Yes it should be. Changes made with proper modifications. 

L89, 90 What are the experimental hypotheses based on? It's not completely consistent with the results of others research (line 62-66). Maybe you can change it to questions.

[Response] No, it meets results of studies in 62-66. Dilution induced by blue light results from promoted biomass accumulation without enough nutrient supply, but that by green light was the result of insufficient nutrient supply. However, we know that we did not reveal enough background to link current hypotheses to core meanings of background. We revised the hypotheses to meet the meaning on paper. 

L149 Only one tanked pot of seedlings received 15N pulsing? One in four? What is the number of total replications of seedlings received 15N pulsing?

[Response] Only the four potted seedlings in one tank of a growing floor received 15N pulse leaving the other four receiving just water. This part is unclear. So, it is revised. Totally, 108 seedlings received pulse and the other 108 ones received water. Either pulsed or controlled seedlings were arranged in 3 replicates as three shelves for one spectrum. 

L210-210 Paragraph spacing is different from before

[Response] Changes made. 

L231 Please hang the first line

[Response] Changes made. 

L270 Describe results that are significantly different firstly, rather than no significant

[Response] Changes made.

L310 But there was a significant influence of light on aboveground biomass in table 2

[Response] acorns grew underground which was not affected by the aboveground biomass. 

L 276-297: I suggest modifying these phrases. First describe what you found and then the possible explanation with citations, rather than a lot of citations. You can check that along all discussion.

[Response] Number of citations will be reduced with additions of more explanations that will help further diluting the density of citations. This will be made throughout the discussion. 

Table 2 I suggest marking the P-values less than 0.05 with asterisks or letters

[Response] Thanks and asterisks will be used. 

7. PLOS authors have the option to publish the peer review history of their article (what does this mean?). If published, this will include your full peer review and any attached files.

Do you want your identity to be public for this peer review? For information about this choice, including consent withdrawal, please see our Privacy Policy.

Reviewer #1: No

Reviewer #2: No

Reviewer #3: No

---

## [Decision Letter · Decision Letter 2]

22 Jun 2021

PONE-D-20-37547R2

Effects of light spectra and 15N pulses on growth, leaf morphology, physiology, and internal nitrogen cycling in Quercus variabilis Blume seedlings

PLOS ONE

Dear Dr. He,

Thank you for submitting your manuscript to PLOS ONE. After careful consideration, we feel that it has merit but does not fully meet PLOS ONE’s publication criteria as it currently stands. Therefore, we invite you to submit a revised version of the manuscript that addresses the points raised during the review process.

We look forward to receiving your revised manuscript.

Kind regards,

Xiao Guo, Ph.D.

Academic Editor

PLOS ONE

Journal Requirements:

Reviewers' comments:

Reviewer's Responses to Questions

**Comments to the Author**

1. If the authors have adequately addressed your comments raised in a previous round of review and you feel that this manuscript is now acceptable for publication, you may indicate that here to bypass the “Comments to the Author” section, enter your conflict of interest statement in the “Confidential to Editor” section, and submit your "Accept" recommendation.

Reviewer #2: All comments have been addressed

Reviewer #3: All comments have been addressed

2. Is the manuscript technically sound, and do the data support the conclusions?

Reviewer #2: Yes

Reviewer #3: Yes

3. Has the statistical analysis been performed appropriately and rigorously? 

Reviewer #2: Yes

Reviewer #3: Yes

4. Have the authors made all data underlying the findings in their manuscript fully available?

Reviewer #2: Yes

Reviewer #3: Yes

5. Is the manuscript presented in an intelligible fashion and written in standard English?

Reviewer #2: Yes

Reviewer #3: Yes

6. Review Comments to the Author

Reviewer #2: (No Response)

Reviewer #3: I find the paper significantly improved. The paper is an interesting contribution to the ecology of coastal wetlands. I still pointed out some other minor corrections:

L64-72 More references about the response of Quercus to different spectra are needed.

L77 Please make “the other two spectra” more clear.

L277-278 Does this sentence mean that individual ontogeny differences are not caused by genetics but phenotypic plasticity?

Table 2 You did not revise at all.

7. PLOS authors have the option to publish the peer review history of their article (what does this mean?). If published, this will include your full peer review and any attached files.

Reviewer #2: No

Reviewer #3: No

---

## [Author Response · Author response to Decision Letter 2]

22 Jun 2021

PONE-D-20-37547R2

Effects of light spectra and 15N pulses on growth, leaf morphology, physiology, and internal nitrogen cycling in Quercus variabilis Blume seedlings

PLOS ONE

Dear Dr. He,

Thank you for submitting your manuscript to PLOS ONE. After careful consideration, we feel that it has merit but does not fully meet PLOS ONE’s publication criteria as it currently stands. Therefore, we invite you to submit a revised version of the manuscript that addresses the points raised during the review process.

We look forward to receiving your revised manuscript.

Kind regards,

Xiao Guo, Ph.D.

Academic Editor

PLOS ONE

Journal Requirements:

[Response] About the bibliography, please believe that none of our citations will be missed either in the text or in reference list. This is because we cite studies by EndNote and all citations will match records in the list accordingly if any changes occurred. 

Reviewers' comments:

Reviewer's Responses to Questions

Comments to the Author

1. If the authors have adequately addressed your comments raised in a previous round of review and you feel that this manuscript is now acceptable for publication, you may indicate that here to bypass the “Comments to the Author” section, enter your conflict of interest statement in the “Confidential to Editor” section, and submit your "Accept" recommendation.

Reviewer #2: All comments have been addressed

Reviewer #3: All comments have been addressed

2. Is the manuscript technically sound, and do the data support the conclusions?

Reviewer #2: Yes

Reviewer #3: Yes

3. Has the statistical analysis been performed appropriately and rigorously?

Reviewer #2: Yes

Reviewer #3: Yes

4. Have the authors made all data underlying the findings in their manuscript fully available?

Reviewer #2: Yes

Reviewer #3: Yes

5. Is the manuscript presented in an intelligible fashion and written in standard English?

Reviewer #2: Yes

Reviewer #3: Yes

6. Review Comments to the Author

Reviewer #2: (No Response)

Reviewer #3: I find the paper significantly improved. The paper is an interesting contribution to the ecology of coastal wetlands. I still pointed out some other minor corrections:

L64-72 More references about the response of Quercus to different spectra are needed.

[Response] We added relevant studies in the early body of this part that introduced the light spectra on oak seedlings. 

L77 Please make “the other two spectra” more clear.

[Response] They are red- and green-lights. Revised. 

L277-278 Does this sentence mean that individual ontogeny differences are not caused by genetics but phenotypic plasticity?

[Response] Although this sentence cannot be understood by extreme meaning, we are afraid, at least for oak seedlings, yes. However, according to the study on ontogeny of oak seedlings by Joshua Sloan and Douglas Jacobs, it was the ontogeny difference that impacted the phenotypic plasticity that imposed a greater impact than just genetic distinction. 

Table 2 You did not revise at all.

[Response] As suggested, we added asterisks for significant level P values in Table 2. Sorry for late revision. 

7. PLOS authors have the option to publish the peer review history of their article (what does this mean?). If published, this will include your full peer review and any attached files.

Do you want your identity to be public for this peer review? For information about this choice, including consent withdrawal, please see our Privacy Policy.

Reviewer #2: No

Reviewer #3: No

---

## [Editor Report · Decision Letter 3]

1 Jul 2021

Effects of light spectra and 15N pulses on growth, leaf morphology, physiology, and internal nitrogen cycling in Quercus variabilis Blume seedlings

PONE-D-20-37547R3

Dear Dr. He,

We’re pleased to inform you that your manuscript has been judged scientifically suitable for publication and will be formally accepted for publication once it meets all outstanding technical requirements.

Kind regards,

Xiao Guo, Ph.D.

Academic Editor

PLOS ONE

Additional Editor Comments (optional):

All comments have been addressed.
---

## [Editor Report · Acceptance letter]

5 Jul 2021

PONE-D-20-37547R3 

Effects of light spectra and ^15^N pulses on growth, leaf morphology, physiology, and internal nitrogen cycling in *Quercus variabilis* Blume seedlings 

Dear Dr. He:

I'm pleased to inform you that your manuscript has been deemed suitable for publication in PLOS ONE. Congratulations! Your manuscript is now with our production department. 

Kind regards, 

on behalf of

Dr. Xiao Guo 

Academic Editor

PLOS ONE